

**Technical note: The recovery rate of free particulate**
**organic matter is strongly reduced by conventional**
**density fractionation of soil samples**
Frederick Büks[1]
[1]Chair of Soil Science, Dept. of Ecology, Technische Universität Berlin, 10587 Berlin, Germany
*Correspondence to:* Frederick Büks (frederick.bueks@tu-berlin.de)
**Abstract.** Ultrasonication combined with density fractionation (USD) is a method widely
used to quantify soil organic matter pools. A selective fractionation of free particulate
organic matter (fPOM) is crucial to avoid co-extraction of retained fPOM along with
occluded particulate organic matter (oPOM). In the present work, artificial fPOM was
extracted from two mineral matrices, sandy and loamy, after applying different approaches
for merging sample and dense medium. It could be shown that pouring the dense solution
to the mineral matrices leads to low recovery, whereas trickling the sample into the
solution, rotating after fill-up or applying a minimal and defined amount of ultrasound to
swirl up the sample causes nearly full recovery of the artificial fPOM. Applied to natural
soils, the results confirmed the low extraction rate of the fill-up approach. Moreover, it was
possible to show that the rotational approach results in only a slightly increased extraction
rate, whereas the ultrasound approach leads to a release of oPOM into the fPOM fraction
due to disruption of soil macro-aggregates. The trickle approach appears to be the most
appropriate way among the tested methods to achieve complete and selective extraction
of fPOM from natural soil samples.
**Introduction**
In soils, particulate organic matter (POM) occurs free (fPOM) as well as occluded within
soil aggregates (oPOM) (Golchin et al., 1994). Both organic matter pools with different
chemical composition, structure and decomposition rates are task of widespread
experimental issues regarding carbon pool balances, soil structural stability or turnover
times (von Lützow et al., 2007; Wagai et al., 2009; Büks and Kaupenjohann, 2016; Graf-
Rosenfellner et al., 2016). A widely used method to separate fPOM and oPOM is
ultrasonication combined with density-fractionation (USD) (Kaiser and Berhe, 2014). Both
POM fractions are thereby determined indirectly by quantification of the operational non-
aggregated particulate free light fraction (fLF) and the occluded light fraction within soil
aggregates (oLF) (Golchin et al., 1994; Büks and Kaupenjohann, 2016). The congruence
between light fractions and actual POM pools is reduced by low recovery rates and the
carryover between the pools as recently shown for oPOM and mineral-associated organic
matter (MOM) (Büks et al., 2021). A sharp separation without cross-contamination
between the measured pools is therefore necessary.





This work focused on the separation of fPOM and oPOM, driven by two observations: A
pre-experiment following the specifications given below for the extraction of POM from soil
samples showed a separation of 28.7±3.1 mg fPOM when the density fractionation
solution was added to the soil sample, but 44.8±7.4 mg when the sample was gently
trickled through the solution (n=3, t-test, p<0.05). The first of the two approaches is the
original and now commonly used in soil science (e.g. Golchin et al., 1994; Cerli et al.,
2012; Schrumpf et al., 2013). However, many works applying the USD do not specify the
method of bringing soil sample and dense solution together. Due to the very different
performance of both approaches shown, the aim of this work was to compare methods in
order to identify such with most accurate separation of fPOM and oPOM.
**Material and methods**
*The simple scenario: Extraction of LD-PE particles from mineral matrices*
In a first experiment, two simple model soils were prepared from a mineral matrix of
calcinated fine sand (89.7 % sand, 9.3 % silt, 1.0 % clay) and a calcinated clayey silt
(8.7 % sand, 69.7 % silt, 21.6 % clay), each amended with 1 wt% of weathered low-density
polyethylene made from cryo-milled film (LD-PE, weathered 96 h at 1000 W m$^{-2}$, 38°C and
50 % RH following DIN EN ISO 4892-2/3, $x_{10\%}$=246 μm, $x_{50\%}$=435 μm, $x_{90\%}$=691 μm,
ρ=0.92 g cm$^{-3}$) as a   well-defined fPOM representative. This setting provides low
physicochemical interaction between mineral and organic particles, that allowed for
focusing on artifacts caused by mechanical reasons such as sedimentation behavior and
impeded flotation. The textures of the two mineral matrices represent different
sedimentation rates, likely affecting the recovery rate of the LD-PE.
Four scenarios with each six replicates of 20 g soil sample and 100 ml 1.6 g cm$^{-3}$ dense
sodium polytungstate solution (SPT) in 200 ml centrifuge bottles were tested: One in which
the soil samples were gently *filled up* with solution, one in which the soil samples were
*trickled* into the solution, one in which the flasks were gently *rotated* horizontally 3x with
20 rpm to unhitch the sedimented soil matrix from the bottom of the flask, and one that
was agitated by ultrasonication (Branson© Sonifier 250, sonotrode diameter 13 mm,
frequency 40 kHz, immersion depth 15 mm, power output 52.06±1.67 J s$^{-1}$) until the
sediment was completely swirled up (*pre-sonicated*). The respective time of sonication
($t_{min}$) was determined to be 7.0±1.3 sec for the sandy and 34.0±1.9 sec for the loamy soil
(*see Supplements*). The corresponding energy densities $w_{min}$ were calculated following
North (1976) and amounted to 3.0±0.5 J ml$^{-1}$ and 14.7±0.8 J ml$^{-1}$, respectively.
In order to extract the POM, samples were centrifuged at 3,500 G for 26 min. The floated
LD-PE was collected by use of a water-jet vacuum pump and cleaned with deionized water
to remove remaining SPT salt by use of a 0.45 μm cellulose acetate membrane-filter until
the electrical conductivity of the filtrate fell below 50 μS cm$^{-1}$. The extracted LD-PE was
then flushed off the filter with deionized water into aluminum bottles, frozen at -20°C,
lyophilized and finally weighed to determine the recovery rate.





*The complex scenario: Extraction of POM from natural soils*
In a second experiment, two topsoil samples, sandy (89.7 % sand, 9.3 % silt, 1.0 % clay)
and loamy (25.5 % sand, 55.9 % silt, 18.7 % clay), were air-dried and sieved to receive
aggregates of 250 to 2000 μm in diameter. In six-fold replication, 20 g of soil aggregates
were gently adjusted via spray to a water content of 200 mg g$^{-1}$ dry soil, low enough to
avoid aggregates sticking to each other or to the flask, and incubated for 2 weeks at 20 °C
in the dark. After the removal of shoots, soil samples and SPT solution were merged
following the four approaches and the fPOM was extracted in the same manner given
above. Subsequently, the samples were refilled to 100 ml of SPT per flask and treated by
application of w=50 J ml$^{-1}$ in the *fill-up*, *trickle* and *rotate* scenarios as well as w=50 J ml$^{-1}$-
$w_{min}$ to the *pre-sonicated* variant. Afterwards the oPOM was extracted as above, followed
by centrifugation, collection, cleaning, freezing, lyophilization and quantification by
weighing. Finally, all POM samples were ground, dried at 105°C and the amount of organic
carbon was determined using an Elementar Vario EL III CNS Analyzer.
**Results**
*Recovery rates from mineral matrices*
The results show that *fill-up*, the commonly used method, provided by far the lowest
recovery rate in both the sandy and clayey mineral matrix (68.3±9.0 % and 58.9±13.7 % of
the applied LD-PE, respectively). In contrast, *trickle*, *rotate* and *pre-sonicated* have
similarly high recovery rates ranging from 90.4±5.8 % to 98.2±1.1 % across all samples
(Fig. 1).

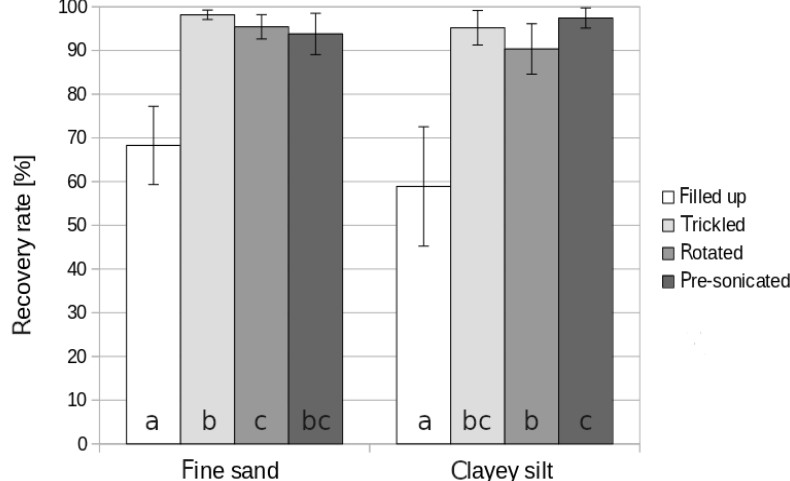

**Figure 1:** Recovery rates of fPOM (weathered LD-PE) from mineral matrices after fractionation with 1.6 g cm$^{-3}$ dense SPT solution using different approaches (n=6, t-test, p<0.05). Small letters indicate Tukey's characters.




*Recovery rate and POM quality in natural soil samples*


The application of all four approaches to aggregates of the loamy natural soil showed, that
the *filled up* samples released by far the lowest amount of fPOM and percentage of total
SOC, followed by the *rotated* and clearly excelled several times over by the *trickled* and
*pre-sonicated* variant (Table 1). Unlike the other fPOM, the *pre-sonicated* fraction
contained large amounts of dark fine material in addition to coarse POM. This comes
along with the lowest C:N ratio, slightly reduced compared to the other fPOMs, and an
increased C:N ratio in the residuum. The yield of the *pre-sonicated* oPOM fraction was
strongly reduced compared to the other variants and showed the release of almost
exclusively fine material. This is in contrast to *fill-up*, *trickle* and *rotate*, which had similar
appearance with traces of coarse material. In sum, the *trickled* sample had the largest
release of ΣPOM=fPOM+oPOM, followed by the *rotated* samples.

**Tab. 1:** Soil organic matter (SOM) release of a loamy topsoil after different approaches for merging sample and dense medium. fPOM refers to the free particulate organic matter floating after application of 0 J ml$^{-1}$, oPOM to the occluded particulate organic matter released after application of 50 J ml$^{-1}$ (*in case of the variant with minimum ultrasonication 15 and 35 J ml$^{-1}$, respectively). $C_{tot}$ refers to the total carbon content of each organic matter fraction including the residuum. ± refers to the standard deviation. Small superscripts are Tukey's characters ($p<0.05$).

| Loamy soil | filled up | trickled | rotated | pre-sonicated* |
|---|---|---|---|---|
| fPOM |  | | | |
| oPOM | | | | |
| fPOM (g kg$^{-1}$ dry soil) | 5.44±1.67 [a] | 14.94±1.96 [b] | 9.68±0.95 [c] | 15.64±1.69 [b] |
| oPOM (g kg$^{-1}$ dry soil) | 13.42±1.43 [a] | 12.39±2.19 [a] | 12.82±0.87 [a] | 1.96±1.67 [b] |
| ΣPOM (g kg$^{-1}$ dry soil) | 18.86±3.10 [a] | 27.33±4.15 [b] | 22.20±1.82 [c] | 17.60±3.36 [a] |
| fPOM (% $C_{tot}$) | 5.18±1.46 [a] | 13.78±3.01 [b] | 8.62±0.88 [c] | 17.13±1.16 [d] |
| oPOM (% $C_{tot}$) | 17.31±5.00 [a] | 13.54±1.21 [a] | 13.88±0.83 [a] | 1.86±1.65 [b] |
| residuum (% $C_{tot}$) | 77.50±5.76 [abc] | 72.68±2.20 [a] | 77.50±0.76 [b] | 81.01±1.16 [c] |
| fPOM (C:N ratio) | 26.05±0.93 [ab] | 25.34±1.55 [ac] | 27.62±1.55 [b] | 24.15±0.61 [c] |
| oPOM (C:N ratio) | 22.00±0.89 [a] | 20.07±0.29 [b] | 20.52±0.78 [b] | 20.23±5.45 [ab] |
| residuum (C:N ratio) | 12.15±0.27 [a] | 11.79±0.32 [a] | 12.01±0.35 [a] | 12.53±0.20b [b] |



Similar to the loamy soil, the *filled up* sandy soil samples showed the smallest amount of
extracted fPOM followed by the *rotated* ones (Table 2). The *pre-sonicated* and *trickled*
samples released the highest amount of fPOM significantly increased by about 93 %
compared to the *filled up* samples. This pattern appears similarly with SOC. The release of
oPOM from *pre-sonicated* samples was reduced compared to the *filled up*, *trickled* and
*rotated* samples. In sum, the *filled up* samples released the smallest and the *trickled*
sample the highest amount of ΣPOM.
In contrast to the rougher treated loamy samples (15 J ml$^{-1}$), *pre-sonication* of sandy
samples with 3 J ml$^{-1}$ did not cause any additional release of fine material within the fPOM
fraction. There were no significant differences of the C:N ratio between all variants, and all
fPOM fractions showed a very similar appearance. On the other hand, the oPOM fractions
of the *filled up* samples and, to a lesser extent, the *rotated* samples showed an increased
number of coarse particles similar to those found within the fPOM fraction, whereas the
*pre-sonicated* oPOM fraction contained nearly no coarse material. This comes along with
the occurrence of the highest oPOM C:N ratio in the *filled up* samples and the lowest in the
*pre-sonicated* and *trickled* samples. Similar to the loamy samples, the residual C:N ratios
in all sandy soil variants are low compared to the fPOM and oPOM fractions, and showed
the highest values in the *filled up* and *rotated* variants.

**Tab. 2:** Soil organic matter (SOM) release of a sandy topsoil after different approaches for merging sample and dense medium. fPOM refers to the free particulate organic matter floating after application of 0 J ml$^{-1}$, oPOM to the occluded particulate organic matter released after application of 50 J ml$^{-1}$ (*in case of the variant with minimum ultrasonication 3 and 47 J ml$^{-1}$, respectively). $C_{tot}$ refers to the total carbon content of each organic matter fraction including the residuum. ± refers to the standard deviation. Small superscripts are Tukey's characters ($p < 0.05$).

| Sandy soil | fill-up | trickle | rotate | US* |
|---|---|---|---|---|
| fPOM | | | | |
| oPOM | | | | |
| fPOM (g kg$^{-1}$ dry soil) | 6.86±1.37 [a] | 13.52±2.97 [b] | 9.37±1.79 [c] | 12.97±2.81 [b] |
| oPOM (g kg$^{-1}$ dry soil) | 8.84±0.20 [a] | 7.28±2.12 [ab] | 7.81±1.65 [a] | 5.73±1.33 [b] |
| ΣPOM (g kg$^{-1}$ dry soil) | 15.70±1.57 [a] | 20.80±5.09 [b] | 17.18±3.44 [a] | 18.70±4.14 [ab] |
| fPOM (% $C_{tot}$) | 4.68±0.91 [a] | 8.97±1.62 [b] | 6.67±1.36 [c] | 11.46±2.16 [d] |
| oPOM (% $C_{tot}$) | 8.23±1.67 [a] | 6.37±2.10 [ab] | 7.65±1.69 [a] | 4.75±1.39 [b] |





| residuum (% $C_{tot}$) | 87.10±2.26 [a] | 84.66±2.33 [ab] | 85.68±1.16 [ab] | 68.79±2.84 [b] |
|---|---|---|---|---|
| fPOM (C:N ratio) | 20.84±1.35 [a] | 19.46±0.96 [a] | 19.88±1.01 [a] | 20.81±1.87 [a] |
| oPOM (C:N ratio) | 18.94±0.47 [a] | 16.02±0.66 [b] | 17.39±1.09 [c] | 15.45±0.77 [b] |
| residuum (C:N ratio) | 8.76±0.21 [a] | 9.40±0.48 [b] | 8.75±0.15 [a] | 9.13±0.52 [ab] |

## Discussion

It was possible to show significant differences in the extraction performance of the different
approaches. As demonstrated in the first experiment, the recovery rate of LD-PE particles
from sandy and loamy mineral matrices is strongly reduced by use of the conventional *fill-*
*up* method. This implies that filling the dense solution on top the soil sample causes parts
of the fPOM to be buried under the mineral matrix. Consequently, it is suggested that the
*fill-up* approach is not an adequate method to avoid incomplete extraction of fPOM. The
retained fPOM will be in turn found within the oPOM fraction leading to both
underestimation of the fPOM and overestimation of the oPOM fraction. The other
approaches were shown to have similar extraction performance in terms of non-occluded,
weakly interacting LD-PE particles within a solely mineral matrix.
However, during extraction of POM natural soils provide additional interference between
SOM and the mineral phase such as by physiochemical interaction of surfaces, biofilm
formation, density gradients of organic matter as well as occlusion within soil aggregates.
The second experiment was therefore performed with samples of aggregates from sandy
and loamy natural soils.
Similar to the first experiment, in both the sandy and loamy soil the extracted amount of
fPOM was strongly reduced in the *filled up* variant, but also in the *rotated* variant,
compared to the two others. Since the fPOM of the sandy soil shows a similar C:N ratio
and composition of coarse particles across all approaches, the fPOM of all sandy soil
variants can be considered free of (fine particulate) oPOM. In turn, the oPOM fractions of
the *filled up* and *rotated* variant contain more coarse material and have a significantly
higher C:N ratio compared to the others. In consequence, the *trickling* and *pre-sonication*
caused less cross-contamination and are, thus, both considered yielding and sharp
methods to extract fPOM from sandy soil samples. Due to its higher total POM yield,
*trickling* is to be preferred over *pre-sonication* for the quantification of soil carbon pools.
In contrast to the sandy soil, the fPOM of *pre-sonicated* loamy sample contains significant
amounts of fine material and a decreased C:N ratio. This artifact can be explained by the
application of mechanical stress through the use of $w_{min}$ to swirl up the soil sample. The
ultrasound led to the disruption of macro-aggregates and the release of a more strongly
degraded soil organic matter fraction. As shown by Wagai et al. (2009) and Cerli et al.
(2012), such fractions can have in some cases a lower C:N ratio. The effect is missing in
the sandy soil samples, which were treated with only 3 J ml$^{-1}$, but appears at 15 J ml$^{-1}$ with
loamy soils. Following Kaiser and Berhe (2014), the applied energy is well below ultrasonic
levels that have been reported to disperse soil aggregates, but may still break down very



weak macro-aggregates. In contrast, data of North (1979) and Golchin et al. (1994) point
out, that even low dispersive energies <10 J g$^{-1}$ already lead to a strong release of clay
particles from aggregates of a clayey soil.
In addition, the oPOM yield of the *pre-sonicated* variant is strongly reduced coming along
with an increased SOC content of the residuum. This effect did not appear with plastic
particles in the first experiment and might be related to ultrasonic comminution of POM as
described in Büks et al. (2021). Although *pre-sonication* provides the highest fPOM yield in
loamy soils, this method is not recommended due to the low total POM yield as well as
aggregate disruption and cross-contamination between POM pools. The greatest release
of total POM by far is achieved using the *trickle* approach, which caused no signs of cross
contamination.
Based on the performance of the four approaches (Table 3), the following general
recommendations are made on their use. The commonly used *fill-up* method is greatly
affected by its very low fPOM recovery and fPOM artifacts within the oPOM fraction.
*Rotating* shows characteristics similar to the *fill-up* approach. It allows a higher, but still
insufficient POM recovery from natural soil samples, while applying an undefined amount
of mechanical stress to aggregates. Together with the *trickle* approach, *pre-sonication*
shows the highest fPOM yield, might be effective when applied to sandy soils, but causes
cross-contamination and low oPOM yield with loamy soils. The *trickling* method, in turn,
avoids mechanical agitation, has high recovery of fPOM combined with the highest total
POM yield and hardly shows any visible nor measurable cross-contamination. Suitable for
a wide range of water contents, it might be, however, inadequate for the application on
very moist or saturated field-fresh or pre-incubated samples that adhere to the sampling
container in such way that it is difficult to transfer without mechanical stress e.g. by use of
a spoon.

**Table 3:** Performance of the four different approaches (fill-up, trickling, rotation and pre-sonication). oPOM recovery is called unknown, if the the oPOM fraction is contaminated with fPOM material.

| | | recovery | | cross-contamination | |
|---|---|---|---|---|---|
| | | **fPOM** | **oPOM** | **oPOM in fPOM** | **fPOM in oPOM** |
| sandy | filled up | low | unknown | no | yes |
| | trickled | high | high | no | n |
| | rotated | medium | unknown | no | yes |
| | pre-sonicated | high | low | no | n |
| loamy | filled up | low | unknown | no | yes |
| | trickled | high | high | no | no |
| | rotated | medium | unknown | no | yes |
| | pre-sonicated | high | low | yes | no |

Based on the findings, a modification of the traditional approach is recommended, that
includes gentle *trickling* of field fresh or pre-incubated samples with water contents below

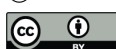



field capacity into the density separation solution instead of adding the solution to the
sample. This avoids burying significant parts of the fPOM under the mineral phase during
the extraction of the fLF, which is then co-extracted along with the oPOM in the following
step.
**Conclusion**
The complete and selective extraction of POM fractions with ultrasonication/density
fractionation (USD) is an important step of SOM pool quantification. It is shown, that the
common approach (filling dense solution onto the soil sample) causes strongly decreased
recovery of fPOM and a cross-contamination with fPOM and oPOM between the free and
occluded light fractions. This causes the misquantification of both fractions and might lead
to the underestimation of the labile and an overestimation of the intermediate soil carbon
pool. In addition to a number of unsuitable alternatives, *trickling* (the soil sample into the
dense solution) has been identified as best approach with high fPOM recovery and low
cross-contamination. As a consequence, a modification of common USD practice by
replacing the *fill-up* with the *trickling* procedure is suggested.
**Acknowledgements**
Many thanks to our technical assistants Maike Mai, Sabine Dumke, Mahboobeh
Behmaneshfard, Sandra Kühn und Sabine Rautenberg, who helped me here and there
with their strong arms and thoughtful contributions.
**Author contribution**
Frederick Büks developed and conducted the experiment, analyzed the data and prepared
the manuscript.
**Competing interests**
The author declares that he has no conflict of interest.





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
