# Peer review of "The recovery rate of free particulate organic matter"

_Biogeosciences, 2022_

## Author Response (AR1)

Dear Daniel Wasner,
many thanks for your detailed, mindful and kind comments. It really has helped me to see some points, that could improve the article. In the following, I want to explain how I propose to adjust the manuscript. My replies to your comments are marked with purple [numbers].

Best regards,
Dr. Frederick Büks

**General Comments:**

In my understanding, this study further shows that for the sake of reproducibility, fractionation studies should describe this method step explicitly.
[1] Thank you, I added to line 203: „For the sake of reproducibility, fractionation studies should describe the way of merging sample and dense solution explicitly."
* * *
**Specific Comments:**

Please clarify (…). (...) I therefore think that this manuscript could be strengthened by a clearer discussion of the potential extent of the problem, by pointing out which commonly used protocols do not mention any sort of SPT/soil mixing, and which protocols feature some sort of mixing and may thus "dodge" the largest bias.

You are right. I changed the narrative of the manuscript in a way, that I accent the diversity of non-mixing and mixing approaches leading to the need of testing their outcome to avoid fPOM-oPOM contamination. For this I:
[2] changed the title to "The recovery rate of free particulate organic matter from soil samples is strongly affected by the method of density fractionation"
[3] added "(1)" in line 37.
[4] replaced Lines 41-46 with: "(2) The treatments of mixing soil sample and dense solution prior to the extraction of fPOM apply a wide range of mechanical stress ranging from non-mixing (Büks and Kaupenjohann, 2016) to swaying ( Rosenfellner et al., 2016), gentle inversion (Golchin et al., 1994), swirling (Cerli et al., 2012), shaking (Schrumpf et al., 2014) and ultrasonic pre-treatment (Don et al., 2009). Due to the very different performances of the above approaches and the diversity of applied treatments, the aim of this work is to compare methods with different underlying principles of mixing in order to identify those with most accurate separation of fPOM and oPOM."
[5] deleted/replaced "conventional", "commonly used" and similar misleading wordings in lines 92, 130, 174, 196 and 202.
[6] added the additional literature to the reference section.
* * *
A comment related to this point: Maybe consider renaming the treatment "filled-up" to something like "unmixed". (...)
[7] Thank you very much for that recommendation. Done.
* * *
Similarly, please consider modifying L13 to "[…] pouring the dense solution to the mineral matrices without mixing […]" to avoid misunderstandings. In L173f this should also be specified.
[8] Both done.
* * *
I suggest to better use the word "treatment" throughout the manuscript instead of "scenario" and "variant" when you refer to the four experimental treatments which you tested.
[9] Done.
* * *
Please add a brief method section on the statistics (…). (…) Also, for Tables 1 & 2 please specify which test was used, and whether the compact letter display of the Tukey post-hoc test (presumably) shows significant differences for each variable between the four treatments (i.e. each line in the table), or within each variable group (i.e. all CN values within a treatment etc).
The ANOVA-Test was conducted for both soils separately.
[10] I added the following paragraph to line 90: "Recovery rates from mineral matrices, fPOM, oPOM and ΣPOM release, proportions of total carbon in fPOM, oPOM and residuum fraction as well as corresponding C:N ratios were compared for all soil matrices separately by one-way analysis of variance (ANOVA) and Tukey test."
[11] I added „… and mark significant differences between the treatments of the (...) soil …" to tables 1 and 2.
* * *
Please somewhere in methods or discussion clearly and explicitly state how you assess the issue of "cross-contamination" in your experiment. I guess you base your conclusions on cross-contamination on observations of CN ratio or on differences in recovered masses in the various fractions, but clarity of the MS could be greatly increased if this was explicitly introduced.
[12] I added „This indicates the input of parts of the coarser fPOM fraction, that has a higher C:N ratio." to line 145 and „and less coarse" to line 157.
* * *
L 38f: Which protocol did the author follow in this pre-experiment? Please specify. Was there some sort of mixing in the first approach?
[13] I added "without mixing" to line 40.
* * *
L 41: "through" is a bit confusing, maybe "into" would be more accurate?
[14] Done.
* * *
L 54: Is there a reference or particular rationale why LD-PE after the described treatment is a good or common (?) proxy for fPOM?
It is within the density and size range of POM and has widely non-reactive surfaces, which reduces surface interactions with the mineral phase in the first experiment, that focuses on simple physical retention of the POM.
* * *
L 62: The treatment "rotated" is not crystal clear to me (but would be important to clarify I think). Does this mean that the vials were tilted three times from a vertical position to a horizontal position (i.e. tilted by 90°)?
[15] I added "… tilted by 90° and axially *rotated* …"
* * *
L 65: I like that the sonication is thoroughly described!
You are welcome.
* * *
L 75: What is lyophilizing, and why is it done?
[16] Lyophilization is the technical term for freeze-drying. I put that in brackets.
* * *
L77f: I find it a bit confusing that the soil material used for the experiments (dried soil sieved to 250 – 2000 um) is referred to as "aggregates". Technically this could still contain sand and fPOM aside from aggregates, since there was no density separation involved?
That's correct, but the two samples I used visually appeared to be composed exclusively of aggregates. For that reason, any influence of unbound primary particles can be assumed insignificant.
* * *
L 81: Out of interest, why were the soils preincubated?
In air-dried soils, (1) microbial biofilms, which are influencing aggregate stability by binding soil particles, are dehydrated and less stable, and (2) very low water content results in precipitation of different salts, if solubility is overshot, which causes cementation. Both leads to a shift in aggregate stability and is avoided by pre-incubation.
* * *
L 84ff: So after "Subsequently …", the entire protocol starting with SPT density fractionation (including the sample+SPT mixing according to the four treatments) was conducted again on what was left as the "heavy fraction" after the first density fractionation? If so, this could maybe be clarified more directly to facilitate quick understanding.
[17] I rephrased the sentence in lines 84-86: "Subsequently, all samples were refilled to 100 ml of SPT per flask, and were equally treated by application of $w = 50$ J ml$^{-1}$ with exception of the *pre-sonicated* treatment, that received $w = 50$ J ml$^{-1}$-$w_{min}$."
* * *
L 89: Did you also measure N for the CN ratio this way?
[18] Yes, thank you. I added „…" to line 89.
* * *
Tables 1 & 2: The numbers presented as "% Ctot" are introduced in the caption as "[…] total carbon content of each organic matter fraction […]". This description in the caption sounds to me as if it was a fraction-specific C content. However, the numbers (e.g. POM 5 %, values up to 80 %) seem to rather be the percentage of total SOC that is contained in the respective fraction (which would make it the same as you introduce in L99 if I understand correctly)? Please introduce more clearly what the values in the table are, and whether in L99 you also refer to Ctot. Aside from that, I like the use of the pictures.
[19] You are right. And thank you very much. I replaced the old description by "$C_{tot}$ refers to the percentage of total SOC contained in each fraction." in both tables.
* * *
Throughout discussion: What exactly is meant with "coarse" (e.g. L 146) and "fine" (e.g. L 154) material? Please specify or define somewhere in the text if the use of these terms is consistent throughout the manuscript.

[20] coarse is related to "less degraded", fine to "more decomposed".
* * *
L 97: The header refers to POM quality, but the notion of "quality" is raised nowhere else. Does this refer to the CN ratios?

[21] Thank you. Replaced by "Recovery rate and characteristics of POM in natural soil samples".
* * *
L166f: Can you briefly explain this idea of "ultrasonic comminution of POM" further, so that the meaning is clear without looking up the source literature?

[22] I added "… leading to strong sorption of the fine particle fraction to the mineral matrix …" in line 167.
* * *
Also, reduced oPOM yield of pre-sonicated treatment in the artificial matrix with LD-PE is not really possible because that artificial soil did not contain aggregates?

That's correct. In the first experiment, I tested the recovery of only the (artifical) fPOM fraction. It is not intended as an approach to test any fate of oPOM.
* * *
L197: This sentence seems to imply that you also observed oPOM in the fPOM fraction of the unmixed treatment (in addition to fPOM in oPOM fraction). That doesn't seem to be the case according to table 3, maybe consider rewording?

[23] Thank you. Fixed by adding "and rotated" in line 196 and "contamination of the occluded light fractions with fPOM" in line 197.
* * *
L199: What is meant with the "intermediate soil carbon pool"?

It is distinguished between a labile, intermediate and stable carbon pool. The respective C turnover rates are related to the degree of decomposition and the location of SOM within the soil structure. The intermediate pool comprises the POM bound within soil aggregates, whereas the stable pool is build by mineral-adsorbed humic substance in micro-aggregates.
* * *
I find the comprehensive provision of raw data in the supplementary material very good!

Thank you!
* * *
**Technical Comments:**

The names of the treatments vary a bit through the MS, sometimes in present tense and sometimes in past tense (e.g. "trickle", "trickled"…) Maybe consider being consistent. At least I would propose to homogenize the nomenclature between the column headers of Tables 1 & 2.

[24] Thank you very much. Done.
* * *
L 8: I wouldn't say USD is a method to "quantify" SOC pools, but rather to separate them
[25] Yes, of course.
* * *
L39: The value after +/- is standard deviation?
[26] I added "± standard deviation" to line 41.
* * *
L 53: What is RH?
[27] Replaced by "relative humidity".
* * *
L 99: "amount of fPOM"... does amount refer to mass (g kg-1)?
[28] I replaced "amount" by "mass".
* * *
L 101f: This sentence is a bit confusing; is it supposed to mean that unlike the fPOM in other treatments, fPOM in the pre-sonicated soils had dark fine material?
[29] Yes. I replaced the sentence by "Unlike the other fPOMs, the fPOM of the *pre-sonicated* treatment has significant amounts of dark fine material."
* * *
L 120: It is a bit unclear what the "increased" refers to. Increased relative to the trickled treatment?
[30] I added "… compared to the other treatments. These particles appeared to be …" to line 121.
* * *
L 139ff: Please provide some reference for the mentioned processes.
[31] I rearranged the paragraph to emphasize, that the statement is not about particular experiments, that found these interactions while extracting POM, but about general physicochemical interactions in soils, that have to be taken into account (and added references).
* * *
Figure 1: What are the error bars?
[32] I added "Error bars refer to standard deviation."

Table 3: Twice "the" in caption, "n" in right column supposed to be "no"?
[33] Done.

Dear referee #2,
thank you very much for your helpful comments. In the following, I want to describe how to adapt to your recommendations. My replies to your comments are marked with green [numbers].

Best regards,

Dr. Frederick Büks

line 41-43 There is also rather gentle approaches with a slow capilary saturation of the soil material. (e.g. Angst, G., Messinger, J., Greiner, M., Häusler, W., Hertel, D., Kirfel, K., Kögel-Knabner, I., Leuschner, C., Rethemeyer, J., and Mueller, C. W. (2018). Soil organic carbon stocks in topsoil and subsoil controlled by parent material, carbon input in the rhizosphere, and microbial-derived compounds. Soil Biology and Biochemistry 122, 19-30.)
Thank you very much. Capillary saturation is an important method to avoid slacking of soil aggregates and might be a suitable way to add dense solution. The recommended reference, however, uses shaking and is therefor an example of the "mixing" group.
* * *
line 59 Was the SPT density tested prior to the experiment? How are differences between the usually used 1.6 and 1.8 g/m^2 affect the outcome of the presented test?
We did not tested a variation of SPT density. Based on the comprehensive work of Cerli et al. (2012, https://doi.org/10.1016/j.geoderma.2011.10.009), who showed 1.6 g/cm³ to be the concentration that extracts the most POM without co-extracting parts of the denser mineral phase, we chose 1.6 g/cm³ for our experiment.
* * *
line 62 Please provide a schematic to illustrate the procedure and thus clarify the actual technical setup.
The protocol of the USD method is properly illustrated by Kaiser and Berhe (2014, https://doi.org/10.1002/jpln.201300339). We hope, that our text description is sufficient to understand the protocol, especially as technical notes are only allowed to include max. 3 tables/figures. However, if wished by the editor, we could add a schematic.
* * *
line 70 floating instead of floated
[34] Thanks. Done
* * *
line 77-89 In general as soon as physical force is used, there is a breakup of aggregates and thus a possible release of oPOM that is then recovered as fPOM. What are the measures to avoid such disruption of oPOM-mineral associations to obtain a clean fPOM fraction?
You're right. Dry sieving of soils to harvest soil aggregates is of course a source of mechanical stress, but applied to all treatments in the same way in order to gain aggregates for this methodical experiment. This step should be avoided when quantifying carbon pools in practice, because (1) only using aggregates warps the carbon picture and (2) the reduction of mechanical stress is crucial in application of all methods of density fractionation of soils. The different principles of merging soil sample and dense solution tested in this work have each a different potential of mechanical stress. However, there is

always a dilemma between the need of treating a sample for measurement on one hand and avoiding artifacts on the other.
* * *
line 79 Please give rational for the use of this aggregate size class.
Aggregates have to be >250µm (operational definition of macro-aggregates) and small enough to fit into the flasks and to provide a sufficiantly high amount of aggregates in each sample. Furthermore, our sandy soil only contained aggregates in the <2mm size class. For similar treatment, we chose a similar size class when testing the loamy soil.
* * *
line 82 Which kind of shoots? Did you seed plants, or the ones from the soil seed bank?
[35] The aggregates are collected in the field and not sterilized to avoid influence on aggregate stability. Under such conditions, it is very normal, that some seeds germinate in a few samples. We removed them as they get visible. We added "… of randomly germinated seeds ..." to line 82.
* * *
line 92-96 If the respective method aims at separating fPOM/fLF and oPOM/oLF, a gentle first step is crucial, and thus an avoidance of the release of oPOM which gets recovered as part of the fPOM. Therefore, the approaches need to be presented in a more nuanced way to clarify the aim of the respective OM separation procedures.
This experiment with mineral matrix and artificial POM did not contain any oPOM. It aims to estimate recovery rates of fPOM without interference of oPOM.
* * *
Table 1 The total C seem to be calculated as a sum to the total bulk soil C. To really be able to evaluate the quality of the separation, the actual C content of each fraction is needed, especially in order to demonstrate the good separation between POM and MAOM/minerals.
$C_{tot}$ is calculated as sum of the C in each fraction, which comprises all C except very little amounts of DOC. The values are displayed in %, which is possible, as only samples of the same soil are compared. Additionally, specific mass data are listed in the supplements. Please also see [19].
* * *
line 160-164 Depending on the soil material, even light shaking/vibration of the beakers containing the soil SPT suspension can lead to aggregate disruption.
That assumption led us to not mix our samples in Büks and Kaupenjohann (2016), but is not provided by our data here (even in the sandy soil), except in the pre-sonicating approach.
* * *
line 171-172 It is hard to evaluate cross-contamination solely based on C contents and C/N ratios.
Table 3 As there is no baseline with a known fPOM and oPOM content, it is hard to tell something about recovery. For this assessment one method would have to be set as a baseline. For the cross contamination assessment, sole C content and C/N ratio are not sufficient to assess this. Also here, a clear baseline would be needed.
We used the first experiment (mineral matrix, artificial POM, no aggregates) to identify approaches with maximum recovery rate of fPOM. Then, we used these methods as a

"baseline" to classify additional fPOM/oPOM amounts as contamination from other pools. We used visual and C:N measurements to proof this assumption, as there is a well known difference in the composition and appearance of fPOM litter and strongly decomposed oPOM. Using natural soil aggregates with in principle unknown SOM pools, this is a suitable approach.
* * *
line 187-192 Also for air dried samples a gentle capilarry saturation does work sufficiently, but an assessment has to be made for every different soil type that is used. Thus, a general remark, as every soil comprises completely different properties, also often simply due to very different POM/MAOM ratios, an adaptation of the fractionation approach has to be made to every new batch of samples. This might mean trickling for one set of samples, stirring for another etc.

Do you have any evidence for that last assumption and maybe a mechanistic explanation for such seemingly random pattern? From our point of view, soils with different composition and thus binding pattern provide different structural stability with values all on a linear scale of mechanical robustness. In consequence, every avoided source of mechanical stress reduces the contamination of pools from other pools. However, work on how binding patterns are differently affected by sources of mechanical stress and therefore cause need for different extraction schemes would be important research.
* * *
line 195 Not just to quantify SOM pools, but also to assess their properties etc. Thus, very often the fractionation is just meant to obtain fractions for further experiments.

[36] We added "and the assessment of their properties" to line 195.
* * *
line 195-200 ...if done in a specific way." Thus, also the proposed trickling method might, if used in a different approach as done by the author, lead to different results. I know it is not really satisfying, but fractionating soils is something that has a human factor in, and also needs adaptation in every lab and for every soil. Thus, the trickling method can be one approach to minimize fPOM/oPOM cross contamination, but only if used in a considerate way...and it is one way besides others.

[37] See penultimate comment. We added "However, mechanical stress patterns might affect different soils with different intensities making other treatments more suitable, which should be considered in upcoming experiments." to line 203.

Reply to the associate editor

Dear Jens-Arne Subke,
thank you very much for your comments. My replies to your comments are marked with orange [numbers].

Best regards,

Dr. Frederick Büks

Line 25: I'm not sure I fully understand the sentence, and particularly what you refer to as "task" and "issues" here. I suggest changing to: "… decomposition rates are the subject of widespread experimental investigations into carbon pool balances…".
[38] Done.

Lines 37 and 45: I propose use of the present tense when referring to your current work. So I suggest: "This work focuses on…" (line 37) and "… the aim of this work is to…" (line 45). Use of past tense when referring to methods and pre-experiment works fine.
[39] Done.

Response to Referee 1 Comments (Daniel Wasner): On comment [4], I suggest a couple of edits in the revised text: "dense solutions" (rather than "solution"); further: "a wide range of mechanical stresses, ranging from …"; "the aim of this work is to…" (as above); "in order to identify those with…".
[40] Done.

Referee comment for line 54 (between [14] and [15]): Your response makes sense, but I agree with the referee that a reference to support the use of LD-PE is appropriate here, or, if that's not achievable, that you add an explanation to justify its use along the lines of your response to referee).
[41] I added to Line 54: "The LD-PE is considered a feasible analogue of natural POM, as it provides a similar range of density and particle size as well as widely non-reactive surfaces, which reduces surface interactions with the mineral phase. This setting ..."

For comment [20]: Please ensure that this interpretation of "coarse" and "fine" in terms of decomposition state is clear in the revised manuscript.
[42] Done.

Response to Referee 2 Comments: I support the addition of a schematic to illustrate the procedure.
[43] Done.